# Peer-coaching interventions for stroke survivors - what works and how: A scoping review

**Yichao Chen**[1], **Paul Victor Patinadan**[1], **Muhammad Amin Shaik**[1],
**Hui Ling Michelle Chiang**[2], **Geraldine Tan-Ho**[1], **Farrah Adystyaning Dewanti**[1],
**Melanie Hui Ru Chng**[1], **Andy Hau Yan Ho**[1,3,4]*

1 Division of Psychology, School of Social Sciences, College of Humanities, Arts and Social Sciences, Nanyang Technological University, Singapore, 2 Discipline of English, School of Humanities, College of Humanities, Arts and Social Sciences, Nanyang Technological University, Singapore, 3 Lee Kong Chian School of Medicine, Nanyang Technological University, Singapore, 4 Palliative Care Centre for Excellence in Research and Education, Singapore

☯ These authors contributed equally to this work.
¤ Current address: Institute of Humanistic Medicine, National Healthcare Group College, Singapore
* andyhyho@ntu.edu.sg

## Abstract

### Background

Peer-led interventions show potential in supporting stroke survivors' recovery, but are described using inconsistent terms and definitions in the current literature.

### Aims

Adopting "post-stroke peer-coaching intervention" as the overarching term, this study aims to synthesise the characteristics and outcomes of existing interventions for stroke survivors to develop a standardised definition and a consolidated summary of findings.

### Summary of review

In this scoping review, we searched 6 databases to identify relevant studies from peer-reviewed journal articles published between January 1993 and October 2025. Data were extracted and analysed regarding intervention definitions, characteristics, and outcomes. The search identified 6609 records, and 8 articles were included, which involved 7 post-stroke peer-coaching interventions. An overall inconsistency was observed across intervention definitions and characteristics. Based on common elements across existing interventions, this study developed an integrated definition, describing post-stroke peer-coaching interventions as a time-limited, patient-centred type of psychosocial and psycho-educational intervention that is ideally developed through participatory action research approaches, and delivers informational, emotional, and appraisal support with application of experiential expertise under the

**Data availability statement:** All relevant data are within the manuscript and its Supporting Information files.

**Funding:** AHYH received financial support from SG Enable & Tote Board, Singapore (ELI Grant Call number 3: Living Independently) (Ref no.: 2022/GC03/18). URL of the funder website: https://www.eli-grant.sg The funder was not involved in the process of study design, data collection and analysis, decision to publish, or preparation of the manuscript.

**Competing interests:** The authors have declared that no competing interests exist.

guidance of healthcare professionals. The analysis also revealed an inconsistency in intervention outcomes.

## Conclusion

Current post-stroke peer-coaching interventions demonstrated inconsistency across definitions, characteristics, and outcomes. To address the inconsistency, this review established a definition that outlines foundational conceptual parameters of the intervention. This proposed definition can serve as a standardised framework to inform the development of future interventions and ensure the provision of systematic support to individuals with stroke.

## Introduction

As the second leading cause of death globally, stroke can engender complex consequences on survivors' psychosocial well-being, such as degraded social participation and development of mental disorders [1,2]. However, supporting post-stroke psychosocial recovery has yet to become a foundational component of current health systems, which often adhere to the biomedical model of disease and prioritise the restoration of physical functions [3]. Therefore, supplementary support is crucial for addressing the wide spectrum of post-stroke needs. In this context, peer-led interventions have emerged as a promising approach, as they foster mutual empowerment between supporters and recipients with a low resource demand, compared with conventional interventions led by health professionals [4]. Preliminary evidence has demonstrated the benefits of peer-led interventions on stroke survivors' psychological well-being, including improved self-efficacy and reduced mental distress [5,6]. A recent review of post-stroke interventions involving peer support also reported significant, positive effects on individuals' emotional regulation and daily activity management [7].

Peer-led interventions can be categorised into various types based on the employed technique that structures the intervention dynamics and the peer-to-peer relationship [8]. Among these, peer-coaching, characterised by a lateral and performance-oriented relationship between individuals with shared experiences is a technique commonly applied for health promotion [8–10]. However, the benefits of post-stroke peer-coaching interventions remain inconclusive due to the absence of a standardised definition to describe and ground such interventions, which results in terminological ambiguity and operational variations. The current literature adopted different terms and inconsistent definitions to describe peer-coaching interventions for stroke survivors. A recent study coined "peer-befriending intervention" and emphasised friendship-based care as the key supportive approach [6]; however, another study used "peer support intervention" with a definition that highlights multiple types of support for recovery facilitation [11]. With such conceptual inconsistency, existing interventions vary widely in content and practice [7]. The heterogeneity in intervention design and operationalisation compromises finding comparability and limits the practical utility of the evidence.

To the authors' knowledge, no previous study has systematically standardised the term and definition of "post-stroke peer-coaching intervention". Such standardisation is essential for establishing a unified framework that guides intervention design and upholds intervention evaluation. To address this conceptual gap, the current study conducted a scoping review, which is a type of review that systematically maps and synthesises existing literature with diverse study designs to summarise available evidence, identify knowledge gaps, or clarify concepts; this scoping review aimed to synthesise existing post-stroke peer-coaching interventions and develop an intervention definition based on the current literature. To enable a systematic literature search, the definition of peer-coaching outlined by Ljungberg and colleagues, which has been widely applied in neurological rehabilitation, was adopted as the working definition to identify targeted studies [12]. Based on this working definition, a post-stroke peer-coaching intervention involves a relationship between individuals both with stroke experience, wherein the experienced individual provides needed support to the less experienced individual. For terminological consistency, the term "peer-coaching intervention" was used throughout this study to address all interventions that align with the working definition, irrespective of the original terminology. Within this framework, individuals providing support are referred to as "peer coaches", those receiving support as "intervention recipients", and the support process as "peer-coaching."

To achieve the research objectives, the current study was guided by the following research questions: i) how does current literature define post-stroke peer-coaching interventions; ii) what are the characteristics of current post-stroke peer-coaching interventions; iii) what are the impacts of current post-stroke peer-coaching interventions.

## Methods

### Study design

The current study employed a scoping review to analyse current research that studied post-stroke peer-coaching interventions. A scoping review was considered the most appropriate study design, since this study focused on definition clarification and evidence mapping. The current scoping review was guided by the five-step approach outlined by Arksey and O'Malley [13], and followed a preregistered protocol (CRD42023478057). This study was reported following the preferred reporting items for systematic review and meta-analyses extension for scoping review (PRISMA-ScR) guideline and the preferred reporting checklist [14].

### Search strategy and eligibility criteria

This study employed the PICOS (problem /intervention /comparison /outcome /setting) and the PICo (population /phenomenon of interest /context) frameworks to develop the search strategy, which is presented in S1 Table [15]. Using the search strategy, a systematic literature search was conducted to identify studies across 6 major databases – Medline, PubMed, Embase, CINAHL, Cochrane Library, and PsycINFO. The literature search included peer-reviewed journal articles published in English between January 1993 and October 2025, given the emergence of peer-led interventions for chronic conditions in the mid-1990s [16,17]. Articles were included if the study employed a quantitative, qualitative, or mixed-method design; the study focused on individuals aged 18 or above living with stroke; the study focused on a peer-coaching intervention that aimed to support stroke survivors and was consistent with the working definition; and the study focused on the development or evaluation of the peer-coaching intervention. Articles were excluded if the intervention was inconsistent with the working definition or if the study solely focused on intervention effects on individuals other than intervention recipients.

### Study selection

All retrieved articles were imported onto the Covidence platform and checked for duplicates. Following the eligibility criteria, two reviewers, MAS and YC, assessed the titles and abstracts of articles independently, and articles deemed relevant by both reviewers entered the full-text screening. After assessing full texts, articles that both reviewers agreed to select

were included. Reference lists of included articles were reviewed to determine if any relevant articles were missed. In cases with conflicts, the third reviewer, PVP, intervened and made the final inclusion/exclusion decision.

## Quality assessment

The methodological quality of included studies was assessed using the Quality Appraisal for Diverse Studies (QuADS) tool [18]. The QuADS tool is a revised version of the Quality Assessment Tool for Studies with Diverse Designs (QATSDD), which has been widely applied by healthcare researchers to assess the quality of studies with varying research designs [18,19]. The QuADS tool addressed limitations of the QATSDD and demonstrated substantial reliability and validity in quality assessment [18]. The tool involves 13 items rated on a 4-point scale, and a higher score indicates greater methodological rigour. The quality assessment was conducted by authors, YC and FAD, independently. Cases with disagreement were resolved by discussion and consensus in consultation with the research team.

## Data extraction and analysis

The lead author YC extracted and tabulated data from the included studies, using a standardised form designed for this review. The form specified the information for extraction, including author, year of publication, country, study design, aim of study, sample, intervention definition, intervention characteristics, and intervention outcomes. Intervention characteristics encompassed elements that reflected similarities and differences across interventions, including development process, component, theoretical basis, treatment window, peer coach recruitment criteria, peer coach training, and mode of delivery. Intervention outcomes included outcomes of intervention recipients reported in quantitative studies and key themes of intervention recipients' experience reported in qualitative studies. The extracted data were reviewed by the second author, PVP, to ensure study rigour. Then the charted data underwent the analysis process, in which the data were iteratively collated, compared, and synthesised to identify similarities and differences in intervention definition, intervention characteristics, and intervention outcomes across the included studies.

# Results

The initial search identified 6609 articles and, with 358 duplicates removed, 6251 articles proceeded to the title/abstract screening. With titles and abstracts assessed, 6220 articles were considered irrelevant based on the eligibility criteria. The remaining 31 articles entered the full-text screening. Fourteen articles were excluded as no full-text was published; nine articles were excluded as the interventions were inconsistent with the working definition, among which 5 interventions did not involve stroke survivors as the main support providers, and 4 interventions concentrated on the formation of support groups without directional support from an experienced stroke survivor, which was delineated in the working definition. Finally, eight articles were included. Fig 1 demonstrates the screening process following the PRISMA-ScR guideline.

## Study characteristics

Characteristics of the included studies are presented in Table 1. Most of the included studies were conducted in Western contexts, whereas one study was conducted in China [5]. As the table indicates, different study designs were employed, including four mixed-method studies, two qualitative studies, and two quantitative studies. In terms of the study aim, six studies focused on intervention evaluation solely, while two studies focused on both intervention development and evaluation [11,20]. Overall, the eight studies involved 1036 participants, most of whom were intervention recipients (N = 825). The number of intervention recipients ranged from 5 to 600 across the studies, and most studies involved intervention recipients fewer than 20. Specific sample size and composition of each included study are presented in Table 2. Sample characteristics in age, gender, ethnicity, and post-stroke time were reported in most studies.

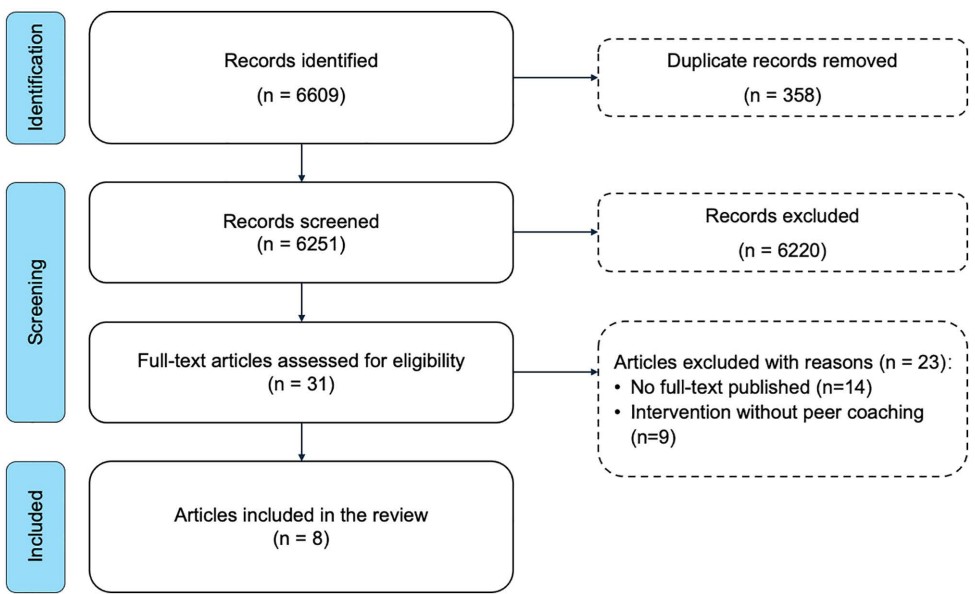

**Fig 1. Flow diagram of the screening process.** Flow diagram of the study selection process, detailing the number of records identified, screened, assessed for eligibility, and included in the scoping review.

**Table 1. Description of the included studies.**

| Author, year (country) | Study design | Study aim | Definition of peer-coaching |
|---|---|---|---|
| Kessler et al. 2014 (Canada) | Qualitative study | Evaluate provided support and intervention effects | Peer coaching provides emotional, instrumental, informational and affirmational social support for people with chronic conditions. |
| Kronish et al. 2014 (US) | Randomised control trial | Assess intervention effectiveness in secondary stroke prevention | Not reported |
| Sadler et al. 2017 (UK) | Mixed-method study | Develop an intervention and evaluate its effectiveness and feasibility | Peer coaching consists of three components: emotional, appraisal and information support. |
| Masterson-Algar et al. 2020 (UK) | Mixed-method study | Co-design and assess an intervention | Peers guide and support people to think about what is possible and to regain a focus on social and leisure activities. Coaching is about support a person to think about a problem and talk through it, rather than providing direct instructions |
| Hilari et al. 2021 (UK) | Mixed-method study | Evaluate feasibility of an intervention | Peer coaches can offer empathy, support, companionship, hope and share experiences and ideas about how to cope. |
| Moss et al. 2022 (UK) | Qualitative study | Explore participants' perceptions of an intervention | Peer coaches offer social, emotional and informational support, to help stroke survivors move forward and develop their own adjustment strategies. |
| Wan et al. 2024 (China) | Randomised control trial | Evaluate intervention effectiveness | Peer coaches can provide emotional, informational, and appraisal support to peer partners. |
| Rose et al. 2024 (Australia) | Mixed-method study | Investigate the feasibility, acceptability, and preliminary efficacy of an intervention | Not reported |

A table that lists key information of the included studies, including author, publication year, country, study design, study aim, and definition of peer-coaching.

**Table 2. Sample Characteristics of the included studies.**

| Study | Sample size | Participants Characteristics | Peer Coach Characteristics |
|---|---|---|---|
| Kessler et al. 2014 | n (intervention recipient) = 16; n (caregiver) = 8; n (peer coach) = 7; n (program coordinator) = 3; n (health professional) = 4 | **Intervention recipients** Mean age: 64.8 years; male: 75%; ischemic stroke: 87.5%; right-hemisphere stroke: 81.3% | n = 7; Mean age: 59.3 years; male: 57.1%; ischemic stroke: 42.9% |
| Kronish et al. 2014 | n = 600 | Mean age: 63 years; female: 59%; non-white: 86%; mean post-stroke time: 1.8 years | Not reported |
| Sadler et al. 2017 | **Qualitative study** n (stroke survivor) = 22: n (caregiver) = 5; n (professional) = 38 **Feasibility study** n (intervention recipient) = 11 | **Qualitative study (stroke survivors)** Age: 62–89 years; post-stroke time: 8–22 months; males: 68.1%; white British: 59% **Feasibility study** Age: 63–87 years old; post-stroke time: 6–23 months; male: 63.6%; white British: 63.6% | Not reported |
| Masterson-Algar et al. 2020 | **Phase I:** n (stroke survivor) = 79 **Phase II:** n (stroke survivor) =1 8; n (caregiver) = 10; n (peer coach) = 5 **Phase III:** n (intervention recipient) =5 | **Phase I:** Male: 62%; mean age: 69.66 years **Phase II (stroke survivors and carers):** Male: 57.14%; mean age: 67.28 years **Phase III:** 3 females and 2 males | n = 5; 3 males and 2 females |
| Hilari et al. 2021 | n = 56 | Mean age: 70.1 years; female: 48.2%; white: 67.9%; black: 25%; average time post stroke: 39.5 days | n = 10; Predominantly female, white, and with experience of ischemic stroke |
| Moss et al. 2022 | n (intervention recipient) = 10; n (caregiver) = 5 | **Intervention recipients** Mean age: 69; female: 50%; black: 50%; white: 40% **Caregivers:** Mean age: 65; female: 80%; white: 60% | Not reported |
| Wan et al. 2024 | n = 120 | Mean age: 67.7 years; male: 53.3%; mean post-stroke time: 35 months | n = 4 |
| Rose et al. 2024 | n (intervention recipient) = 7 n (close others) = 7 | Intervention recipients Mean age: 67 years; male: 86.7% | n = 4 |

A table that lists sample size, participant characteristics, and peer coach characteristics of the included studies.

## Quality assessment

Regarding inter-rater reliability, the two raters reached initial agreement in 66 cases (63.4%), and only in 5 cases (4.8%) was the difference larger than 1, with a scale from 0 to 3 for each item. With a possible maximum score of 39, quality scores of the included studies ranged from 22 to 35 (M = 29.13). Four included studies were rated in the highest quartile, indicating substantial study quality. Most studies clearly stated the research aims (M = 2.63), applied an appropriate study design (M = 2.75) and adopted appropriate strategies for data collection and analysis (M = 2.63). However, there was a general lack of explanation for underpinning concepts (M = 1.88), justification for analytic strategy (M = 1.63), and involvement of stakeholders (M = 1.75) among the studies. The detailed quality assessment results are presented in S2 Table.

## Intervention definition

Definitions that the included studies adopted to describe peer-coaching interventions are presented in Table 1. Most included studies specified an intervention definition to highlight the primary features of post-stroke peer-coaching intervention, whereas two studies introduced no definition [21,22]. Overall, the definitions varied in the content and description manners. Nevertheless, most of the definitions emphasised the types of support provided in the intervention. Commonly indicated support types included emotional support (N = 4), informational support (N = 4), and appraisal/affirmative support (N = 3). Due to description variations, some definitions conveyed similar meanings without an explicit indication of the support types. The definition introduced by Hilari and colleagues implied the focus on emotional support by emphasising

the provision of empathy, companionship, and hope [6]. The definition introduced by Masterson-Algar and colleagues implied the provision of emotional and appraisal support by highlighting "talking through a problem and thinking about what is possible" as the primary support process [20]. However, the definitions were contradictory with the provision of information support: despite the emphasis on knowledge provision in four studies [5,11,23,24], Masterson-Algar and colleagues refuted the provision of direct instructions in peer-coaching and, instead, advocated the prioritisation of emotional and appraisal support [20]. Given this definitional inconsistency, the characteristics of the intervention were subsequently examined to identify commonalities and variations in their design.

## Intervention characteristics

Among the eight studies, seven interventions were involved, with one intervention evaluated by two studies with different methods [6,24]. Besides the included studies, relevant protocol articles were reviewed for additional intervention details [25,26]. The detailed characteristics of involved interventions are presented in S3 Table. Similarities and differences in intervention characteristics are presented below.

## Intervention development process

The development process was explicated for five interventions. One intervention was developed by adopting an existing programme without elaboration of the adaptation process [22], while the other four were developed with involvement of stakeholders, including stroke survivors, caregivers, healthcare professionals, community representatives, outreach workers, and patient educators (S3 Table). One intervention employed the participatory action research (PAR) approach to engage stroke survivors, community educators, and community residents in the intervention co-design [25]; another used a similar participatory method – the active patient, care, and public involvement (PCPI) approach – to co-design the intervention with stroke survivors [6]; the remaining two interventions engaged stakeholders in intervention development without indication of a specific approach [11,20]. In these four interventions, discussions informed by stakeholders' experiences and feedback were carried out to determine key intervention elements.

## Intervention component and theoretical basis

The components of all seven interventions were specified in the studies. Due to varied description manners, themes were generated to summarise the components. Commonly adopted intervention components included information provision (N=5), experience sharing (N=4), action planning (N=3), and problem solving (N=3). Most interventions introduced more than three components jointly to support stroke survivors. To provide individualised support, there were interventions whose components were tailored by recipients [6,20–22]. Only three interventions explicated theoretical frameworks to support the validity of their components. One intervention demonstrated the framework and its application in intervention development [5], whereas the remaining two did not explain the connection between the theoretical foundation and the intervention components [11,20]. Detailed intervention components and theoretical bases of each included intervention are present in S3 Table.

## Treatment window

The treatment windows were specified in most interventions to indicate the post-stroke stage at which interventions were introduced. The specific treatment windows varied across the interventions. One intervention applied an upper limit, focusing on individuals within five years post-stroke [23]. In contrast, two interventions applied a lower limit, one focusing on individuals who had completed hospitalisation and intensive rehabilitation [6,24], and one focusing on those with more than 6 months post-stroke [22]. Furthermore, two interventions applied enclosed treatment windows, one focusing on individuals within 6–24 months post-stroke [11], and one focusing on those post-discharged and within six months post-stroke [20]. Finally, one intervention spanned multiple post-stroke stages, commencing at the acute stage and proceeding after hospitalization [23].

## Peer coach recruitment and training

Only three interventions specified the recruitment criteria of peer coaches, with considerations of post-stroke time length, health conditions, recovery status, and residence [11,22,24,26]. A majority of the interventions offered training to peer coaches, most of which involved didactic sessions focusing on intervention content (N = 4), communication techniques (N = 4), and the peer coach role (N = 3). Besides didactic sessions, there were interventions that introduced interactive training practices, such as discussion, group activity, and coaching session modelling [5,20,23]; three interventions offered training manuals as supplementary material for training facilitation [6,20,22]. Detailed training content of the included interventions is present in S3 Table. In addition to pre-intervention training, ongoing supervision was provided in four interventions to support peer coaches and facilitate intervention delivery [5,20,22,26].

## Mode of delivery

Variances in the mode of delivery were observed across the involved interventions, particularly in format, setting, dosage, and session length (S3 Table). Regarding the format, three interventions adopted the individual-based format, supporting only one recipient in each session; while four interventions adopted the group-based format, supporting multiple recipients collectively in each session. Most of the involved interventions were delivered in-person, while two employed a hybrid format combining in-person and online modes [22,23]. The interventions were delivered in different settings, among which community was the most common one (N = 4).

Among the six interventions that reported the intervention length, five of them lasted for less than 4 months, whereas one lasted for 12 months [23]. All involved interventions reported the total session number: five interventions involved 6 sessions, while two involved 12 sessions [11,22]. The session frequency ranged from 1 session every 2 weeks to 2 sessions per week. In terms of the session length, four interventions introduced sessions each lasting for an hour or shorter; one introduced sessions each lasting for 1.5 hours [21]; two introduced sessions each lasting for 2 hours [5,22].

## Intervention outcome

In the eight included studies, five quantitative investigations and five qualitative inquiries were conducted to evaluate the intervention effects. Detailed intervention outcomes are presented in S4 Table. Among the quantitative investigations, only Wan and colleagues reported significant positive effects on stroke survivors' psychosocial well-being, whereas the remaining did not observe the expected outcomes [5]. The quantitative evaluation conducted by Hilari and colleagues found no significant improvement in general well-being and social well-being, although the intervention group showed a decreased proportion of participants with high distress [6]. Sadler and colleagues detected post-intervention improvement in resilience and QoL, but the increases were marginal and not validated with a significance level [11]. Kronish and colleagues found the intervention effective in controlling the blood pressure level, but no significant effects on the cholesterol level and medication uptake were observed [21].

In qualitative inquiries, intervention recipients consistently reported post-intervention improvement in well-being. Through personal interviews, three studies found perceived benefits to emotional well-being, as recipients reported elevated confidence and hope, and decreased feelings of loneliness [11,23,24]; another study reported the social benefits, as recipients perceived enhanced community connection and participation [22]. Due to a small sample, Masterson-Algar and colleagues collected limited qualitative data to generate a reliable overview of intervention recipients' attitudes and perceptions on the intervention [20].

## Discussion

To the authors' knowledge, the current scoping review is the first study that aims to develop a standardised definition of post-stroke peer-coaching intervention by synthesising the characteristics and outcomes of existing interventions.

Identification of eight related studies reflected a limited body of research that explored this type of intervention. Among the included studies, four studies focused on intervention development and feasibility evaluation, suggesting that post-stroke peer-coaching interventions remain a novel technique at the preliminary stage of intervention development. Despite the inclusion of two randomised controlled trials (RCTs) with large samples [5,21], quasi-experimental and qualitative assessments with small samples were more prevalent in the current literature, which reflects the challenges to apply a rigorous quantitative study design with sufficient participants for intervention evaluation. Based on quality assessment results, although the studies reached a moderate or higher level of quality in general, a majority of them were deficient in conceptual underpinning and research design. Most of the studies provided limited explanation on the concept of "peer coaching", which resulted in the absence of a clearly-defined conceptual framework to uphold the interventions; the employed analytic methods were not justified in most included studies, which undermined the rigor of the research design; although most studies involved stakeholder in the intervention development, limited information was provided to describe the process of involvement.

## Synthesising an intervention definition

The definitions of post-stroke peer-coaching intervention in the current literature were inconsistent. The inconsistency could not be addressed by the working definition, as it failed to capture essential intervention attributes comprehensively. A consensus on the definition can stipulate the intervention features and lay the conceptual foundation for subsequent intervention development and evaluation. To address this gap, the current study developed an integrated definition of post-stroke peer-coaching intervention, grounded in the common elements of existing interventions.

Despite overall inconsistencies, most present studies employed PAR or other participatory paradigms for intervention development. As an integrated strategy to operationalise stakeholder involvement, the PAR paradigm has been widely applied in healthcare intervention development, to identify pathways for health promotion and improve the intervention effectiveness [27]. For post-stroke interventions, this approach engages survivors and other stakeholders to inform specific needs and difficulties, thereby establishing a robust foundation for intervention feasibility and effectiveness. Thus, development with the PAR paradigm is considered as an essential element for post-stroke peer-coaching intervention to ensure patient-centred support. As interventions are co-designed with stakeholders, a prescriptive protocol is needed to guide the content development. Most definitions emphasised the provision of emotional, appraisal, and informational support, originally proposed by Dennis as the pivot of peer-coaching intervention [28]. However, informational support provided by peer coaches carries the risk of misguidance and complications, due to unsecured information quality and accuracy [20,29]. To ensure information fidelity and appropriate delivery, it is crucial that content is reviewed by healthcare professionals and that peer coaches receive sufficient training prior to intervention delivery. By integrating the support types and safeguard measures, the proposed definition outlines parameters for intervention content that enable the delivery of effective and reliable support to stroke survivors.

Building on these foundational elements, the delivery of intervention content emerges as another important feature. In most studies, intervention content was designed to be adapted by recipients, allowing stroke survivors to tailor elements such as goals to achieve, problems to discuss, and activities to participate [6,11,21,24]. This individualisation is considered an essential practice in peer-coaching as it fosters a sense of autonomy and supports the delivery of patient-centred care aligned with the unique needs of stroke survivors [8]. Regarding the delivery of peer-coaching sessions, it was a common practice to implement each session within an hour. Considering the physical and cognitive capabilities of both peer coaches and service users, the session length of one hour or shorter is deemed appropriate to maintain participant engagement and ensure the intervention quality [23].

Besides intervention content and delivery, the treatment window is a critical essential element to consider, given the dynamic trajectory of post-stroke recovery. Most interventions were introduced to stroke survivors after hospital discharge, as the transition from inpatient care to home-based care is often stressful and challenging. After hospitalisation, limited

healthcare support is available to patients and their families, and the care discontinuity can result in distressing uncertainty and recovery disruption [30]. Therefore, stroke survivors at the post-discharge stage are considered as the group with an urgent need for support [6]. On the other hand, peer-coaching is considered impractical at the hospitalisation stage, as the survivors remain physically unstable and mentally unprepared for the support [23]. Hence, the post-discharged stage is considered the optimal treatment window to introduce post-stroke peer-coaching intervention.

Based on the key elements commonly adopted in the present studies, the authors proposed a standardised definition of post-stroke peer-coaching interventions. These interventions are defined as a time-limited, patient-centred type of psychosocial and psycho-educational intervention ideally developed through PAR approaches, delivering informational, emotional, and appraisal support with application of experiential expertise under the guidance of healthcare professionals. It is characterised by a mutually-benefiting relationship in which well-trained and experienced long-time stroke survivors create a safe, non-judgmental, and empathetic environment to provide needed support to aid the holistic recovery of recently post-discharge, less-experienced stroke survivors. By standardising core parameters of intervention design and delivery, this proposed definition provides an overarching conceptualisation that supports the development of consistent post-stroke peer-coaching interventions and enables systematic evaluation of their effectiveness.

## Intervention characteristics

Grounded in the similarities observed across present studies and interventions, the proposed definition outlined conceptual parameters for multiple characteristics of post-stroke peer-coaching interventions, whereas other characteristics remained inconsistent across the existing interventions, including intervention components, peer coach recruitment, and delivery mode. The multi-level heterogeneity impedes systematic investigation of optimal characteristics and intervention effectiveness.

Despite a common focus on emotional, informational, and appraisal support, most interventions introduced a curriculum without practical instructions that elucidate the implementation procedures. One exception was the intervention developed and evaluated by Masterson-Algar and colleagues, which provided detailed information on session operation in the manual [20]. Without detailed instructions, interventions are at risk of inconsistent implementation, undermining the effectiveness and impeding replication. Another issue regarding the curricula is the lack of a theoretical foundation, which leads to concerns over the curriculum efficacy and restricts the comparison between curricula. Furthermore, the criteria and assessment tools for peer coach recruitment were not introduced in most interventions. These elements are necessary to standardise the recruitment procedures, which are methodologically essential for consolidating intervention quality and preventing adverse events.

Regarding intervention delivery, the individual-based format was more common in existing interventions, compared with the group-based format. Although group-based sessions can amplify mutual motivation, their delivery is likely to be disrupted by logistics issues [23]. The individual-based format is considered advantageous, as it provides more spaces for rapport building and intervention individualisation [8,20,24]. Furthermore, most interventions were delivered in-person and in-person delivery is considered more acceptable, as it provides stronger, tangible connections and reduces the feeling of loneliness [8,23,31]. Regarding the intervention setting, although most interventions were delivered in community settings, they are not ideal for every stroke survivor to confide in difficulties and discuss their feelings. Thus, the setting can be flexible with the consideration of individual preferences [20]. The intervention length and the session frequency were highly inconsistent across existing interventions. Therefore, it is difficult to determine an optimal standard for these characteristics.

## Interventions outcomes

Discrepancy in intervention outcomes was observed between the quantitative investigations and qualitative inquiries conducted to evaluate the existing post-stroke peer-coaching interventions. Despite some evidence supporting the intervention effectiveness [5,6,21], most quantitative investigations did not observe the expected outcomes in improving stroke survivors'

well-being or QoL. Although researchers attributed this to external factors, such as the limited sample size [6,11], between-group sociodemographic differences [6], and uncertain clinical practices [21], the results indicated potential deficiencies in the intervention components and implementation and suggested the development of a potent and robust intervention. In comparison, the qualitative evaluations provided consistent evidence that supported the intervention effectiveness. Intervention recipients reported improvement in their emotional well-being and indicated the advantages of peer coaches, who could provide experiential advice and made them feel confident [23]. The result aligns with findings from a review study that concluded experience sharing, vicarious learning, and social comparison are the key features of support provided by peers in post-stroke interventions [4]. Qualitative inquiries in current literature observed the benefits of peer-coaching interventions on stroke survivors' well-being, especially in the psychological aspect, whereas current quantitative investigations did not reach the same conclusion confidently. However, it is noticeable that most of the included qualitative and mixed-method studies indicated a lower level of quality, which undermined the trustworthiness of the concluded findings.

### Limitations of the study

There are limitations to be considered in the current study. Firstly, our search strategy targeted journal articles published solely in English. Related articles published in other languages were excluded from the review and, as a result, interventions developed or implemented in non-English speaking countries were likely to be excluded. As most of the studies reviewed were conducted in developed countries or regions, the generalisability of these findings remains limited. Therefore, caution should be exercised when extrapolating these results to research conducted in different socioeconomic and cultural contexts. In addition, this review only included articles with interventions that were consistent with the working definition. Although this eligibility criterion aimed to exclude interventions without directional support between stroke survivors, there was a possibility that other studies with potential contributions to the conceptualisation were omitted in the current work. Last, since this review focused on conceptual clarification and evidence mapping, no exclusion criteria were applied based on study quality, despite a quality assessment conducted for the reviewed studies. Given the quality variation, readers should consider the quality of individual studies when interpreting and applying their findings.

### Implication for research

Application of a standardised definition is crucial to develop or evaluate a post-stroke peer-coaching intervention, as it can clarify the conceptual parameters of the intervention and minimize terminological confusion. This scoping review can inform future research on intervention development. Building on a synthesis of existing research, the current study proposed an overarching, evidence-based definition to describe the interventions definition, which standardised the internal relationship, the development process, the intervention content, and the treatment window. This definition can serve as a framework and guide future research to develop peer-coaching interventions that facilitate post-stroke recovery systematically. With the concentration on stakeholder involvement and patient-centred care, this definition can also promote the development of interventions that are informed by service stakeholders and provide tailored support in a specific cultural and regional context.

The standardisation in intervention development can also enable rigorous evaluation on intervention effectiveness. In addition, this scoping review can also inform future research on improving research quality. Future research needs to provide a detailed description of intervention characteristics and incorporate rigorous study designs to improve generalizability and validity.

### Conclusion

Peer coaching intervention, a promising approach to support stroke survivors through bridging care discontinuity from hospital discharge to community-based care, has been developed and implemented with inconsistent terms and definitions

in the current literature, which leads to conceptual confusion and precludes a coherent understanding of the intervention. Grounded on a synthesis of existing research, the current study established an overarching, evidence-based definition to describe the intervention, which standardised the internal relationship, the development process, the intervention content, and the treatment window. Narrative syntheses were generated to describe the characteristics and outcomes of the existing interventions. Considering the heterogeneity in both intervention characteristics and outcomes, the definition proposed in this study offers a conceptual foundation for future research to develop post-stroke peer coaching interventions guided by a unified framework, thereby enabling consistent implementation and systematic evaluation.

## Supporting information

**S1 Table. Search strategies.** A table that lists database searches, keywords, and Boolean operators used in the literature search.
(DOCX)

**S2 Table. Assessment results of included studies.** A table that lists the quality assessment results of the included studies, with item-specific scores of each study.
(DOCX)

**S3 Table. Description of post-stroke peer-coaching interventions.** A table that lists detailed characteristics of the included interventions, including development process, component, theoretical basis, treatment window, peer recruitment and training, and mode of delivery.
(DOCX)

**S4 Table. Description of intervention outcomes.** A table that lists the detailed outcomes of quantitative investigations and qualitative inquiries conducted in the included studies.
(DOCX)

**S5 Table. Preferred reporting items for systematic reviews and meta-analyses extension for scoping reviews (PRISMA-ScR) checklist.** A table that describes how the PRISMA-ScR guideline was followed and presented in the current work.
(DOCX)

## Acknowledgments

We would like to acknowledge research members of ARCH lab NTU for their kind assistance on literature search, screening, and manuscript preparation.

## Author contributions

**Conceptualization:** Yichao Chen, Paul Victor Patinadan, Muhammad Amin Shaik, Hui Ling Michelle Chiang, Andy Hau Yan Ho.

**Data curation:** Yichao Chen, Paul Victor Patinadan, Muhammad Amin Shaik.

**Formal analysis:** Yichao Chen, Paul Victor Patinadan.

**Funding acquisition:** Paul Victor Patinadan, Muhammad Amin Shaik, Hui Ling Michelle Chiang, Andy Hau Yan Ho.

**Investigation:** Yichao Chen, Paul Victor Patinadan, Muhammad Amin Shaik.

**Methodology:** Yichao Chen, Paul Victor Patinadan, Muhammad Amin Shaik, Andy Hau Yan Ho.

**Project administration:** Geraldine TAN-HO, Farrah Adystyaning Dewanti, Melanie Hui Ru Chng.

**Resources:** Geraldine TAN-HO, Farrah Adystyaning Dewanti, Melanie Hui Ru Chng.

**Supervision:** Paul Victor Patinadan, Andy Hau Yan Ho.

**Writing – original draft:** Yichao Chen.

**Writing – review & editing:** Yichao Chen, Paul Victor Patinadan, Muhammad Amin Shaik, Hui Ling Michelle Chiang, Andy Hau Yan Ho.

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
