## [Decision Letter · Decision Letter 0]

26 Jan 2026

Dear Dr. Ho,

Thank you for submitting your manuscript to PLOS ONE. After careful consideration, we feel that it has merit but does not fully meet PLOS ONE’s publication criteria as it currently stands. Therefore, we invite you to submit a revised version of the manuscript that addresses the points raised during the review process.

We look forward to receiving your revised manuscript.

Kind regards,

Yilin Jiang

Academic Editor

PLOS One

2. Please remove your figures from within your manuscript file, leaving only the individual TIFF/EPS image files, uploaded separately. These will be automatically included in the reviewers’ PDF.

Reviewers' comments:

Reviewer's Responses to Questions

**Comments to the Author**

1. Is the manuscript technically sound, and do the data support the conclusions?

Reviewer #1: Yes

Reviewer #2: Yes

Reviewer #3: Yes

Reviewer #4: Yes

2. Has the statistical analysis been performed appropriately and rigorously?

Reviewer #1: N/A

Reviewer #2: N/A

Reviewer #3: Yes

Reviewer #4: Yes

3. Have the authors made all data underlying the findings in their manuscript fully available?

Reviewer #1: Yes

Reviewer #2: Yes

Reviewer #3: Yes

Reviewer #4: Yes

4. Is the manuscript presented in an intelligible fashion and written in standard English?

Reviewer #1: Yes

Reviewer #2: Yes

Reviewer #3: Yes

Reviewer #4: Yes

Reviewer #1: I extend my gratitude to the authors for the opportunity to review this manuscript. This scoping review addresses a significant and timely topic concerning peer coaching interventions for stroke survivors. This study undertook a scoping review of peer coaching interventions for stroke survivors. The authors identified inconsistencies in terminology and definitions within the existing literature and aimed to establish a standardized definition by integrating the characteristics and outcomes of these interventions. Utilizing Arksey and O’Malley’s framework and the PRISMA-ScR guidelines, they searched six databases and extracted eight studies. The analysis revealed a lack of consistency in the definitions, characteristics, and outcomes of interventions. Consequently, the authors propose a new integrated definition that incorporates approaches such as Participatory Action Research (PAR) and guidance from healthcare professionals. This study employed appropriate methodologies for a scoping review, including the preregistration of the protocol, adherence to PRISMA-ScR, and selection by multiple reviewers, and the quality of the research was assessed to be high. Notably, in a field characterized by conceptual confusion, organizing the components and proposing a new definition constitute valuable contributions to future interventions. However, the relationship between the working and final proposed definitions requires clarification to prevent confusion among readers. As this could impact the scientific validity of the study, a revision is required.

Major Comments

The authors adopted the “Working definition” from Ljungberg et al. (2011) for their literature search. However, they proposed a “standardised definition” as the final outcome. In the Methods section, it is stated that interventions not matching this working definition were excluded, raising concerns that this procedure may have resulted in the omission of studies that should have contributed to conceptualizing “post-stroke peer-coaching.” In the Discussion section, I recommend addressing, either as a limitation or as an addition to the Methodology, how (or if) filtering with the working definition influenced the final scope setting and generation of the definition.

Minor Comments

Table 1 presents the characteristics of the included studies, and I understand that detailed information on each element designated as the scope—“definitions of interventions,” “characteristics,” and “outcomes”—is provided in Supplemental Materials 3 and 4. However, in the Results section (Intervention Characteristics), I believe that more frequent referencing of Table 1 and the Supplemental Materials would facilitate easier cross-referencing of the main text with the tables for readers.

Reviewer #2: This manuscript presents a well-conducted and timely scoping review addressing an important and underexplored area in stroke rehabilitation—post-stroke peer-coaching interventions. The study is methodologically sound, clearly reported, and aligns well with the stated objectives of conceptual clarification and evidence mapping.

The use of a preregistered protocol, adherence to PRISMA-ScR guidelines, and application of the QuADS tool for methodological appraisal demonstrate strong rigor and transparency. The authors appropriately justify the use of narrative synthesis rather than statistical meta-analysis, given the heterogeneity of study designs, intervention characteristics, and outcome measures. The analytical approach is therefore suitable and robust for a scoping review.

Compliance with the PLOS Data Availability Policy is satisfactory. As this study synthesizes previously published literature, the inclusion of all extracted data, quality appraisal results, and intervention descriptions within the manuscript and supporting information is appropriate and sufficient to support the reported findings.

The manuscript is well structured and written in clear, standard academic English. The arguments are logical, the tables are informative, and the synthesis is presented in an intelligible and accessible manner. The proposed standardized definition of post-stroke peer-coaching interventions is a notable strength and represents a meaningful contribution that can guide future intervention development and evaluation.

Reviewer #3: the manuscript is well structured, methodologically transparent, and thoughtfully discussed. The proposed integrative definition is a key strength and has potential value for future intervention development and evaluation. However, several areas require clarification which include the following

Please clarify how the working definition (based on Ljungberg et al.) may have influenced study inclusion and whether this could have excluded relevant peer-based interventions using alternative conceptualisations.

The decision to include studies from 1993 onward should be briefly justified.

Although quality appraisal is not mandatory for scoping reviews, the implications of including lower-quality studies should be discussed more explicitly when interpreting findings.

The discussion should further address how cultural, healthcare system, and community differences may influence peer-coaching implementation and outcomes.

Reviewer #4: The manuscript entitled “Peer-coaching interventions for stroke survivors – what works and how: a scoping review” is well written, well organized, and demonstrates strong methodological rigor. The study addresses a relevant and timely topic, making a meaningful contribution to the understanding of peer-coaching interventions for stroke survivors.

In the introduction, I suggest minor adjustments to improve the initial understanding of the intervention addressed in the study. At first glance, before a full reading of the manuscript, the approach adopted in the review is not clearly described, which may limit the reader’s understanding of the study’s scope. The objectives are clearly presented; however, a clearer definition of the approach and scope of the review could enhance the clarity and accessibility of this section.

The methodology is carefully developed and appropriately follows the recommended guidelines for scoping reviews. The procedures are described in a transparent and consistent manner. As a suggestion for improvement, I recommend a clearer definition of the inclusion criteria (for example, specifying the age of participants included in the analyzed studies), as well as the exclusion criteria. Nonetheless, the use of methodological protocols is robust and well aligned with the proposed objectives.

The discussion is clear, well structured, and adequately addresses the questions posed by the authors. I suggest that the information presented on page 17, lines 320 to 324, regarding the types of study designs of the included articles, also be incorporated into the results section, thereby strengthening the presentation of the findings. Throughout the text, there are minor punctuation issues that could be addressed to further improve clarity.

The reflections and suggestions presented in the discussion are relevant and insightful. The conclusion is concise and appropriate to the study objectives. The references are current, well selected, and appropriately used; moreover, the recommendation to include age as part of the inclusion criteria could be further justified in this section, considering the existence of newer studies on peer-led interventions across different age groups. Finally, the supplementary materials are appropriate and effectively complement the manuscript.

**Do you want your identity to be public for this peer review?** For information about this choice, including consent withdrawal, please see our For information about this choice, including consent withdrawal, please see our Privacy Policy .

Reviewer #1: No

Reviewer #2:**Yes**

Reviewer #3:**Yes**

Reviewer #4: No

---

## [Author Response · Author response to Decision Letter 1]

11 Feb 2026

The following content can also be found in the "Response to Reviewer" file that has been submitted along the manuscript.

Responses to Comments from Academic Editor

Comment 1: Please ensure that your manuscript meets PLOS ONE's style requirements, including those for file naming.

Comment 2: Please remove your figures from within your manuscript file, leaving only the individual TIFF/EPS image files, uploaded separately. These will be automatically included in the reviewers’ PDF.

Author response: Thank you for this guidance. We have examined the style of the materials and made necessary changes: correcting the format of the manuscript, removing the figure from the manuscript, correcting the file name and format of the figure file, and correcting the file names of supporting information files.

2. Responses to Comments from Reviewer #1

Comment 1: The authors adopted the “Working definition” from Ljungberg et al. (2011) for their literature search. However, they proposed a “standardised definition” as the final outcome. In the Methods section, it is stated that interventions not matching this working definition were excluded, raising concerns that this procedure may have resulted in the omission of studies that should have contributed to conceptualizing “post-stroke peer-coaching.” In the Discussion section, I recommend addressing, either as a limitation or as an addition to the Methodology, how (or if) filtering with the working definition influenced the final scope setting and generation of the definition.

Author response: We thank the reviewer for this critical observation. The working definition was applied to ensure the basic conceptual alignment across the included studies, which focused on interventions that involved stroke survivors as the primary support provider and introduced directional support to the service user. Specifically, it functioned as a systematic filter to differentiate the targeted interventions from other interventions that involved limited peer-based support or introduced support groups with the focus on mutual support between stroke survivors. According to our screening process and record, we believe that the articles that were excluded based on the working definition would not contribute to the development of intervention definition and the current work. However, we acknowledge that this measure of inclusion/exclusion still carries a risk of study omission. Following your guidance, we have indicated the application of the working definition as a limitation of the current work (Page 26). In addition, to address the concern of readers, we have clarified the number of articles that were excluded based on the working definition, as well as the reasons that these articles were excluded (Page 9).

Comment 2: Table 1 presents the characteristics of the included studies, and I understand that detailed information on each element designated as the scope—“definitions of interventions,” “characteristics,” and “outcomes”—is provided in Supplemental Materials 3 and 4. However, in the Results section (Intervention Characteristics), I believe that more frequent referencing of Table 1 and the Supplemental Materials would facilitate easier cross-referencing of the main text with the tables for readers.

Author response: We agree that more frequent reference to the tables will facilitate the reading process. Therefore, we have added several phrases that refer to Table 1 and tables from the supporting information across the result section (Line 188, 190, 197, 220, 222, 254, 274, 298, 304, and 322) .

3. Responses to Comments from Reviewer #2

Author response: We thank the reviewer for the positive assessment of our methodology and for recognising the value of the proposed definition.

4. Responses to Comments from Reviewer #3

Comment 1 : Please clarify how the working definition (based on Ljungberg et al.) may have influenced study inclusion and whether this could have excluded relevant peer-based interventions using alternative conceptualisations.

Author response: We thank the reviewer for the suggestions on clarifying the influence of the working definition, which was also raised by Reviewer#1. Following your guidance, we have clarified the number of articles that were excluded based on the working definition and the reasons that these articles were excluded (Page 9). We believe that the articles that were excluded based on the working definition would not contribute to the development of intervention definition and the current work. However, considering the potential risk of study omission, we have indicated the application of the working definition as a limitation of the current work (Page 26).

Comment 2: The decision to include studies from 1993 onward should be briefly justified.

Author response: We have added this justification on Page 7, Line 128.

Comment 3: Although quality appraisal is not mandatory for scoping reviews, the implications of including lower-quality studies should be discussed more explicitly when interpreting findings.

Author response: We agree that the quality of included studies needs to be further discussed. We have added a more comprehensive summary of study quality in the discussion section (Page 20) and discussed the influence of study quality on the findings (Page 25)

Comment 4: The discussion should further address how cultural, healthcare system, and community differences may influence peer-coaching implementation and outcomes.

Author response: We have added a section that explains how the established definition can be employed to address different needs for support across diverse regions (Page 27), as the definition incorporates participatory action research, which can provide space for intervention tailoring.

4. Responses to Comments from Reviewer #4

Comment 1: In the introduction, I suggest minor adjustments to improve the initial understanding of the intervention addressed in the study. At first glance, before a full reading of the manuscript, the approach adopted in the review is not clearly described, which may limit the reader’s understanding of the study’s scope. The objectives are clearly presented; however, a clearer definition of the approach and scope of the review could enhance the clarity and accessibility of this section.

Author response: We thank the reviewer for the detailed and critical observation. We have added a brief definition to introduce scoping review as the study design in the introduction section (Page 5).

Comment 2: As a suggestion for improvement, I recommend a clearer definition of the inclusion criteria (for example, specifying the age of participants included in the analyzed studies), as well as the exclusion criteria.

Author response: We have added participant age as one of the inclusion criteria and structured the inclusion/exclusion criteria to improve the readability (Page 7).

Comment 3: I suggest that the information presented on page 17, lines 320 to 324, regarding the types of study designs of the included articles, also be incorporated into the results section, thereby strengthening the presentation of the findings.

Author response: We have added the information presented between line 320 and 324 into the results section (Page 10, Line 195 to 197).

Comment 4: Throughout the text, there are minor punctuation issues that could be addressed to further improve clarity

Author response: We have carefully examined the manuscript and corrected punctuation issues throughout the text.

---

## [Decision Letter · Decision Letter 1]

10 Mar 2026

Peer-coaching interventions for stroke survivors - what works and how: A scoping review

PONE-D-25-63712R1

Dear Dr. Ho,

We’re pleased to inform you that your manuscript has been judged scientifically suitable for publication and will be formally accepted for publication once it meets all outstanding technical requirements.

Kind regards,

Yilin Jiang

Academic Editor

PLOS One

Reviewers' comments:

Reviewer's Responses to Questions

**Comments to the Author**

Reviewer #1: All comments have been addressed

Reviewer #4: All comments have been addressed

2. Is the manuscript technically sound, and do the data support the conclusions?

Reviewer #1: Yes

Reviewer #4: Yes

3. Has the statistical analysis been performed appropriately and rigorously?

Reviewer #1: N/A

Reviewer #4: Yes

4. Have the authors made all data underlying the findings in their manuscript fully available?

Reviewer #1: Yes

Reviewer #4: Yes

5. Is the manuscript presented in an intelligible fashion and written in standard English?

Reviewer #1: Yes

Reviewer #4: Yes

Reviewer #1: I have reviewed the revised manuscript and the authors' responses to my comments.

regarding the Major Comment, the authors have successfully clarified the influence of the "working definition" on the study selection process. The addition of detailed exclusion reasons in the Methodology section and the acknowledgment of potential study omission as a limitation in the Discussion section adequately address the concern about the study's scope and validity.

Regarding the Minor Comment, the inclusion of frequent cross-references to Table 1 and the Supplemental Materials throughout the Results section has significantly improved the readability and navigability of the manuscript.

The authors have engaged constructively with all points raised. I believe the manuscript has been improved significantly and is now suitable for publication.

Reviewer #4: The suggested corrections were implemented, contributing to making the article clearer and more fluid to read. The proposed changes were carefully considered, and the revision demonstrates attention in incorporating the recommendations provided. Overall, the care and dedication applied to the revision process are clear, resulting in a text that is more organized, consistent, and easier to understand.

**Do you want your identity to be public for this peer review?** For information about this choice, including consent withdrawal, please see our For information about this choice, including consent withdrawal, please see our Privacy Policy .

Reviewer #1: No

Reviewer #4: No

---

## [Editor Report · Acceptance letter]

PONE-D-25-63712R1

PLOS One

Dear Dr. Ho,

I'm pleased to inform you that your manuscript has been deemed suitable for publication in PLOS One. Congratulations! Your manuscript is now being handed over to our production team.

Kind regards,

on behalf of

Dr. Yilin Jiang

Academic Editor

PLOS One